# Adsorption of Ag (I) Ions at the Zirconium Phosphate/KNO_3_ Aqueous Solution Interface

**DOI:** 10.3390/ma15145050

**Published:** 2022-07-20

**Authors:** Władysław Janusz, Ewa Skwarek, Volodymyr Sydorchuk, Svitlana Khalameida

**Affiliations:** 1Faculty of Chemistry, Maria Curie-Skłodowska University, Maria Curie-Skłodowska Sq. 3, PL-20031 Lublin, Poland; wladyslaw.janusz@poczta.umcs.lublin.pl; 2Institute for Sorption and Problems of Endoecology NAS of Ukraine, Naumova Street 13, 03164 Kyiv, Ukraine; bilychi@ukr.net (V.S.); svkhal@ukr.net (S.K.)

**Keywords:** zirconium phosphate, modification, adsorption, Ag

## Abstract

The paper presented the mechanical (MChT), microwave (MWT), and hydrothermal (HTT) methods of zirconium phosphate samples modification in order to improve its adsorption affinity for the Ag (I) ions. The FTIR studies proved that the modification of both gel and xerogel samples with the ultrasonic microwaves causes an increase in the concentration of phosphate groups on the surface of MWT-modified zirconium phosphate: the isoelectric point pHiep = 2.2–2.9 for these samples against 3.9 for the initial sample and pKa2 values were 4.7–5.6 and 6.3, respectively. As resulting from the Ag^+^ ion adsorption studies, the MWT treatment of zirconium phosphate samples caused the greatest affinity of Ag^+^ ions for the surface of MWT zirconium phosphate. Compared with the initial ZrP sample, the shift of the Ag (I) ion adsorption edge towards lower pH values was observed, e.g., with adsorption of Ag (I) ions from the solution with the initial concentration of 1 µmol/dm^3^ for the initial ZrP sample pH50% = 3.2, while for the sample MWT ZrPxero pH50% = 2.6.

## 1. Introduction

Silver ions are characterized by antibacterial properties as well as low toxicity to humans. Although known from antiquity, the antibacterial properties of Ag^+^ ions, have been appreciated in recent years due to the increasing resistance of bacteria to antibiotics [1]. Owing to this property, silver ions, silver nanoparticles (AgNps), and silver antibacterial agents are used to treat various infections [2,3]. Among the silver antibacterial agents, in addition to the AgNPs deposited on nanofibers, there were obtained other inorganic silver carriers, such as zeolites, phosphates, titanium dioxide, activated carbon, montmorillonite, and MoS_2_ [4,5].

Crystalline zirconium phosphates, having a two-dimensional layered structure, as well as those in the amorphous state exhibit ion exchange properties and are characterized by large thermal, chemical, included in the acidic media, and radiolytic stability [6,7]. It was shown that, owing to these properties, zirconium phosphates can be widely applied in materials chemistry and based on them it is possible to obtain catalysts, drug delivery carriers, or adsorbents for purifying water from pollutants. The studies of zirconium phosphate doping with Ag ions as a result of NaZrP ion exchange proved that the obtained silver zirconium phosphate is characterized by antibacterial properties against *Escherichia coli* and *Staphylococcus aureus* [4]. The antibacterial activity against *Escherichia coli* was demonstrated by the samples of titanium zirconium phosphate on which Ag^+^ ions were adsorbed [8]. Besides, the Ag-containing zirconium phosphate synthesized by the ion-exchange procedure was studied as an effective oxidation catalyst [9,10,11], photocatalyst [12,13,14], and adsorbent for dye removal from aqueous media [15]. Therefore, the investigations of silver cations sorption on zirconium phosphate is important from the point of view of the preparation of these materials. It should be noted that there is only a limited amount of research on this issue [4,8,16]. At the same time, the influence of the porous structure of ZrP on the Ag^+^ sorption has not been studied at all. Thus, modification of zirconium, titanium, and tin phosphates under hydrothermal conditions (using conventional hydrothermal, microwave, and mechanochemical treatments) demonstrates wide possibilities for controlling the parameters of the porous structure, surface structure and, as a result, their adsorption characteristics [7,17,18,19]. Therefore, it is important from the scientific and applied points of view to study the features of these types of treatment for phosphates and the relationship between the physicochemical characteristics, and the adsorption properties of phosphates was established, including in relation to Ag^+^ ions. Based on this, the synthesis of the Ag-containing materials can be improved.

The studies of Ag^+^ ion adsorption using the zirconium phosphate samples will allow determining the conditions for removing ^110m^Ag radioisotope ions from the nuclear liquid waste in order to purify these waters. Another aspect of the research is to obtain a material that, on one hand, will be characterized by great antibacterial activity and, on the other hand, will be chemically and thermally stable.

This paper presented the results of the studies on adsorption of Ag^+^ ions as a function of pH in the amorphous zirconium phosphate samples previously subjected to the hydrothermal, mechanical, and microwave modifications in order to vary the parameters of the porous structure and surface design and its adsorption activity with respect to Ag^+^. The discussion on the activity of the groups on the surface of the modified zirconium phosphate samples was held based on the FTIR spectra.

## 2. Materials and Methods

### 2.1. Adsorbents

The zirconium phosphate wet gel with the moisture content of about 88% was obtained by reacting zirconium chloride with phosphoric acid. Its composition is Zr(HPO_4_)_2_. The details of the synthesis and purification of the sample are described in ref. [17]. Part of the Initial sample was dried at 20 °C for 50 h in order to prepare the xerogel designated Initial. The gradual increase of the temperature to 250 °C resulted in a sample designated as ZrPxero.

The wet gel and dried xerogel were subjected to the hydrothermal, microwave, and mechanochemical treatment (hereinafter designated as HTT, MWT, and MChT) as in ref. [17]. HTT was performed at 150 °C for 3 h using the laboratory autoclave, MWT, at 230 °C for 0.5 h using the “NANO 2000” high-pressure reactor (Plazmatronika, Wrocław, Poland), MChT, at 500 rpm for 0.5 h using the planetary ball mill Pulverisette-7, premium line (Fritsch Gmbh, Idar-Oberstein, Germany). The list of the tested samples and their designation is presented in Table 1. As can be seen, the selected samples differ in their specific surface areas.

### 2.2. Physicochemical Methods of Investigations

The details of the procedure used for physicochemical characteristics determination are described in ref. [17]. In order to measure nitrogen adsorption/desorption based on the obtained adsorption isotherms, the analyzer ASAP 2405N (“Micromeritics Instrument Corp.”, Norcross, GA, USA) was used. The specific surface area and porosity of the tested zirconium phosphate samples were determined; the details of the procedure are discussed in ref. [17]. The FTIR spectrum of the tested samples was recorded with a Nicolet 8700A FTIR spectrometer in the wavelength range 4000–400cm^−1^. All samples were also examined using the diffractometer Philips PW 1830 (Amsterdam, The Netherlands) with CuKα-radiation (λ = 0.15406 nm) for the crystal structure characterization.

Measurements of the of silver ions adsorption at the interface of the zirconium phosphate sample/0.001 mol/dm^3^ KNO_3_ solution interface + Ag^+^ ions with the concentrations of 1, 10, and 100 μmol/dm^3^, respectively, were made by means of the method of radioactivity loss from the solution using a measuring system also in the potentiometric titration. Nitrogen, purified from CO_2_, was passed through the measuring system consisting of a Teflon vessel, propeller stirrer, glass electrode, and calomel electrode. Then, 50 cm^3^ of the electrolyte solution was placed in a Teflon vessel, and then the solution containing the radioactive isotope ^110m^Ag^+^ was added. From the solution prepared in this way, zero samples with the volume of 0.1 cm^3^ were taken, and then the weighed amount of zirconium phosphate precipitate was added. The resulting suspension was vigorously stirred, and at the time of equilibrium, 0.5 cm^3^ samples of the suspension were taken which were then centrifuged. From each of the centrifuged samples, 0.1 cm^3^ of the supernatant solution was taken twice and placed on the filter paper, thus creating sources for radioactivity measurements using the Beckman Gamma 5500B automatic scintillation counter (Brea, CA, USA). The modification procedure changed the surface specific area which affects the adsorption results. Moreover, the amount of the adsorbent was selected so that each adsorbent contacted with the same surface of the Ag^+^ solution (Table 1).

The density of the ion sorption (A_Ag_) on the zirconium phosphate surface was calculated from the changes in radioactivity before and after the adsorption using the formula:$$A_{\mathrm{Ag}}=\frac{c_{0}V}{mS_{w}}\left(1-\frac{N_{r}}{N_{0}}\right)$$
and
$$c_{r}=c_{0}\frac{N_{r}}{N_{0}}$$
where:*c*_0_—the Ag^+^ initial concentration (mol/dm^3^)*V*—the volume of the solution (dm^3^)*m*—the adsorbent mass (g)*S_w_*—the specific surface area (m^2^/g)*N_r_*—the number of counts from the source taken during the adsorption,*N*_0_—the number of counts from the source taken before adsorption,*c_r_*—the Ag^+^ equilibrium concentration (mol/dm^3^)


## 3. Results and Discussion

### 3.1. FTIR Study of the ZrP Samples

As shown in ref. [17], initial precipitated ZrP is X-ray amorphous. Moreover, its phase composition is preserved after the hydrothermal microwave and mechanochemical modifications although some ordering of the structure takes place [18,19]. The FTIR spectra of the obtained initial sample and the samples modified as dried xerogel are shown in Figure 1 while those for the samples modified in the form of wet gel are depicted in Figure 2. The bands in the range of 1000–1100 cm^−1^ assigned to the asymmetric and symmetric vibrations of the PO_4_^3−^ groups as well as those in the range of 400–800 cm^−1^ related to the vibrations of the Zr-O bond [10,11] had the same intensity. Stanghellini et al. analyzed in detail the IR spectra of the samples γ-Zr [O_3_POH]_2_·H_2_O and α-Zr [PO_4_] [O_2_P(OH)_2_]·H_2_O and proved that there are symmetrical and asymmetric vibrations of P-O and P-O-H in the wavenumbers 850–1300 cm^−1^ [20]. Only the MWT causes the broadening of the band associated with the vibrations of the phosphate groups in the range of 1300–850 cm^−1^, and in particular, increases the intensity of the bands related to P-O-H. The MChT treatment of the gel ZrP sample reduces the share of the intensity of the bands in the peak of the phosphate groups. The presented changes indicate that the MWT treatment causes the most significant changes. Besides, the surface structure is changed as a result of MWT and MChT. Thus, the FTIR spectroscopic studies of the initial and modified samples showed a wide band in the spectra in the wavenumber range 3700–2400 cm^−1^ with the maximum around 3450 cm^−1^, which is associated with the vibrations of the water molecules and surface OH groups. The band at the wavenumber 1630 cm^−1^ is attributed to the bending vibrations of the O-H bond in the water molecule [21,22]. The spectra for the initial sample and after HTT of gel and xerogel showed a similar course. On the other hand, the samples after both MWT and MChT were characterized by more intense bands in the range of 3700–2400 cm^−1^ and the band at the wavenumber of 1630 cm^−1^. As can be seen, the intensity of the band at 3450 cm^−1^ increased significantly after MWT and MChT. These bands are related to the presence of the adsorbed H_2_O and surface OH groups. The increase of the intensity of this band for the milled sample (Figure 2) may be due solely to a drastic increase in the specific surface area (Table 2). However, a more significant increase in the intensity of this band for the samples after MWT (Figure 1 and Figure 2) is obviously explained not only by an increase in the specific surface area.

It should be noted that MWT used in this study is the version of HTT. However, the observed effect of microwave treatment on the surface structure (according to the FTIR data) is different from that of the conventional hydrothermal treatment. This may be due to a specific non-thermal effect that manifests itself during MWT [23]. Non-thermal effects are those that are not due to the increase of the material thermal energy.

### 3.2. Electrophoretic Study of the Zirconium Phosphate Samples

The FTIR studies showed the presence of Zr-OH and P-O-H groups in the samples. These groups on the ZrP surface can undergo the following ionization reactions, resulting in a surface charge formation:(1)$$\equiv {\mathrm{ZrOH}}_{2}^{+}↔\equiv {\mathrm{ZrOH}+H}^{+}$$
(2)$$\equiv \mathrm{ZrOH}↔\equiv {\mathrm{ZrO}}^{-}{+H}^{+}$$
(3)$$\equiv \mathrm{POH}↔\equiv {\mathrm{PO}}^{-}{+H}^{+}$$

The adsorption of the proton on the hydroxyl group according to reaction (1) leads to the formation of a positive charge while the dissociation of the proton from the hydroxyl group, reaction (2), or dissociation of the proton from the hydrogen phosphate group, reaction (3), leads to the formation of a negative charge. The charge accumulated on the ZrP surface is the algebraic sum of charges originating from the above-mentioned groups. In the case of a low concentration of the base electrolyte, Sprycha and Szczypa assumed that the surface charge density is approximately equal to the charge density of the diffusion layer which can be calculated from the zeta potential measurements [24]. Assuming further that for pH < pHiep the surface charge is determined by the groups with the positive charge and for pH > pHiep by the negatively charged groups, they proposed a method of calculating the thermodynamic constant equilibria of the ionization reaction of groups with the positive charge K_a1_ and those with the negative charge K_a2_ [24].

Figure 3a,b show the dependence of the zeta potential as a function of the pH of the xerogel and gel ZrP samples, respectively, at 0.001 mol/dm^3^ KNO_3_. The relationships of the zeta potential as a function of the pH of the xero ZrP samples presented in Figure 3a differed significantly from the dependence of the zeta potential versus pH for the Initial sample. Initially the xerogel sample showed a similar course up to pH ~ 3.5, but above this pH value the zeta potential was higher, which indicates a smaller concentration of the negatively charged groups. The ZrP MWT xerogel sample in the range from pH = 2 to pH = 6.5 was characterized by smaller zeta potential values, which indicates greater ionization of the surface groups towards the formation of negatively charged groups. Based on the FTIR spectrum of this sample, it can be assumed that the hydrogen–phosphate groups can be responsible for such a course. The dependence of the zeta potential as a function of pH for the gel-type ZrP samples (Figure 3b) shows that the gel-type sample showed, in the initial pH range, up to pH = 5, lower pH values and higher than the those of ZrP initial sample. On the other hand, the ZrP MWT and ZrP MChT samples up to pH = 7 exhibited smaller zeta potential values, and above pH = 7 the zeta potential vs. pH was not much different from that of the initial sample.

The pH_iep_ values and the negative logarithm of the thermodynamic equilibrium constants pK_a1_ and pK_a2_ calculated from these relationships are presented in Table 2. The pH_iep_ value of the xerogel sample in relation to the pH_iep_ of the Initial sample was slightly higher due to the lower concentration of negatively charged groups; the surface groups were prone to ionization as evidenced by the value of pK_a2_ = 7.2. On the other hand, the surface groups of the Gel, MWT gel, and MChT gel samples were more acidic. The pK_a2_ of the surface groups of these samples was significantly smaller than the pK_a2_ of the Initial sample (Table 2), and the pH_iep_ of these samples was shifted towards lower pH values compared with those of the Initial sample. The functional groups of the xerogel ZrP after HTT had the most acidic character, the pK_a2_ being 1.5 units lower than the pK_a2_ of the initial sample and hence the pH_iep_ of this sample was 2.2.

### 3.3. Adsorption of Ag (I) Ions at the Interface of the ZrP Sample/Solution 0.001 mol/dm^3^ KNO_3_

Zirconium phosphates or titanium phosphates are the adsorbents characterized by the great adsorption affinity for heavy metals [25,26]. It was found that these adsorbents can practically remove Cd (II) and Zn (II) ions from the solutions with a pH greater than 4. Adsorption of heavy metal cations in the pH range from 2 to 4 increases with the increasing pH due to the exchange of protons in the group’s zirconium or titanium phosphate hydroxyl or hydrogen phosphate group. Ag^+^ ions can also adsorb to one hydroxyl or phosphate group releasing a proton. Due to the small pH_iep_ values, ionized (negatively charged) forms are present next to the hydroxyl and hydrogen phosphate groups which, can also adsorb silver cations on the zirconium phosphate surface.

Figure 4, Figure 5 and Figure 6 show the relationship between the adsorption of Ag^+^ cations as a function of pH on various samples of zirconium phosphate with the solution at the initial concentrations of Ag ions of 1, 10, and 100 µmol/dm^3^, respectively. With the increase in pH from pH = 3 to ~6, the adsorption of silver ions increased, and then after exceeding the value pH = 6, a slower increase in the adsorption occurred up to the maximum adsorption, which for the initial concentrations of Ag^+^ ions was equal to 1, 10, 100 μmol/dm^3^, and 2, 20, and 200 µmol/m^2^, respectively. The course of Ag^+^ vs. the pH cations adsorption resembles the adsorption of multivalent cations on metal oxides and is called the adsorption edge. This edge was characterized by the pH value at which 50% of the initial Ag^+^ concentration is adsorbed (pH_50%_) and the pH range in which the adsorption changes from 90% to 10% (ΔpH_90–10%_) [27]. The analysis of the dependence of Ag^+^ cation adsorption as a function of pH showed that the zirconium phosphate samples exposed to MWT are characterized by a shift of the above-mentioned edge towards smaller pH values, which proves a greater adsorption affinity for the Ag^+^ cations compared with the other samples. This effect can be explained by the larger content of P-O-H groups determined from the FTIR spectra.

A convenient method to determine the amount of protons released from the ZrP surface groups as a result of adsorption is the non-stoichiometric cation exchange according to the reaction [28]:(4)$$n\equiv {\mathrm{SOH}+\mathrm{Ag}}^{+}\Leftrightarrow {\left(\equiv {\mathrm{SO}}^{-}\right)}_{n}{\mathrm{Ag}}^{\left(1-n\right)+}{+nH}^{+}$$
where:*n*—the number of surface groups reacting with the silver cation = the number of released H^+^ cations.


Since the adsorption interactions of heavy metal cations with the surface groups are strong and specific, the component of electrostatic interactions can be neglected and the adsorption reaction constant can be described as follows:(5)$$\beta_{n}^{s}=\frac{\left[{\left(\equiv {\mathrm{SO}}^{-}\right)}_{n}{\mathrm{Ag}}^{\left(1-n\right)+}\right]{\left[H^{+}\right]}^{n}}{{\left[\equiv \mathrm{SOH}\right]}^{n}\left[{\mathrm{Ag}}^{+}\right]}$$
where:*β_n_^s^*—the thermodynamic constant of the reaction.


By transforming the equation to a linear form, similarly to refs. [14,15], and then by calculating new variables $\mathrm{lg}\left(\frac{\left[{\left(\equiv {\mathrm{SO}}^{-}\right)}_{n}{\mathrm{Ag}}^{\left(1-n\right)+}\right]}{\left[{\mathrm{Ag}}^{+}\right]}\right)$ and $\mathrm{lg}\left(\frac{\left[\equiv \mathrm{SOH}\right]}{\left[H^{+}\right]}\right)$, and extrapolating the dependencies $\mathrm{lg}\left(\frac{\left[{\left(\equiv {\mathrm{SO}}^{-}\right)}_{n}{\mathrm{Ag}}^{\left(1-n\right)+}\right]}{\left[{\mathrm{Ag}}^{+}\right]}\right)$ as a function of $\mathrm{lg}\left(\frac{\left[\equiv \mathrm{SOH}\right]}{\left[H^{+}\right]}\right)$, the values *β_n_^s^* and *n* can be derived. The obtained values of *β_n_^s^* and *n* are presented in Table 3. As can be seen for all tested concentrations and adsorbents, the number of released protons was smaller than 1, which proves that a significant part of Ag^+^ ions is adsorbed on the negatively charged groups.

After determining the equilibrium constants and the amount of released protons, the pH values at which 50% of the initial concentration of Ag^+^ ions were adsorbed (pH_50%)_ and in the pH range in which the adsorption changes from 90% to 10% (ΔpH_90–10%)_ were calculated. The obtained values of pH50% and ΔpH_90–10%_ are given in Table 3. As can be seen, with the increase of the initial concentration, the pH_50%_ value shifted towards higher pH values; however, in most cases it did not exceed pH = 4. Similarly, the value of ΔpH_90–10%_ exceeded 4 in two cases.

## 4. Conclusions

X-ray amorphous precipitated zirconium phosphate Zr(HPO_4_)_2_ was subjected to the hydrothermal and microwave modifications in the form of wet gel and dried xerogel. The specific surface area of the modified samples was within 195–450 m^2^/g. Their crystal structure was almost the same. On the other hand, the FTIR tests showed stronger bands characteristic of the Zr-OH groups and the appearance of vibrations for various phosphate groups. The presence of these groups caused a shift of pH_iep_ = 3.9 for the Initial ZrP sample to pH_iep_ = 2.2 for the MWT treated Xerogel ZrP sample. The studies of the adsorption of Ag (I) ions from the solutions with the initial concentrations of 1, 10, and 100 μmol/dm^3^ on zirconium phosphates modified with MCHT, MWT, and HTT showed that the xerogel and MWT modified zirconium phosphate gel samples were characterized by high adsorption affinity for Ag (I) ions. Compared with the ZrP initial sample, there was a shift of the Ag (I) ion adsorption edge towards lower pH values, e.g., with adsorption of Ag (I) ions from the solution with the initial concentration of 1 µmol/dm^3^ for the ZrP Initial sample pH_50%_ = 3.2, while for the sample MWT ZrP_xero_ pH_50%_ = 2.6. The effect could be of practical importance in removing the ^110m^Ag radioisotope from the nuclear wastewater.

## Figures and Tables

**Figure 1 materials-15-05050-f001:**
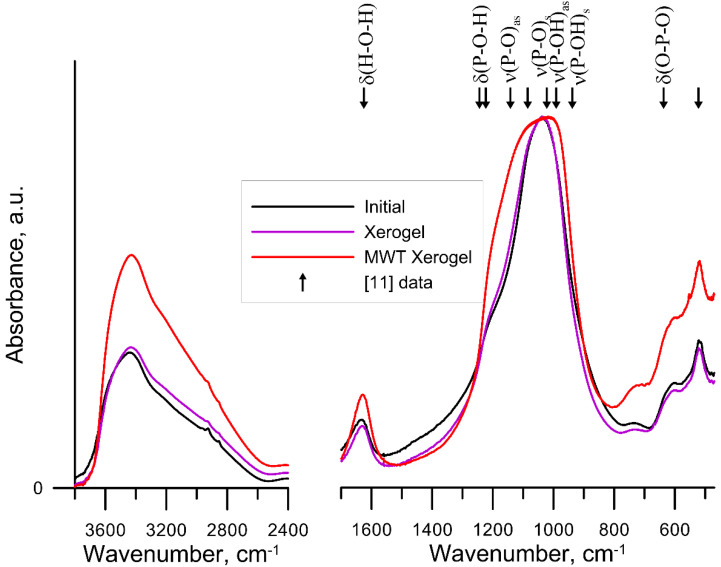
Comparison of FTIR spectra for the initial (black) and after HTT (violet)- and after MWT (red)-treated xerogel samples [11] Xu, Y et al., 2020.

**Figure 2 materials-15-05050-f002:**
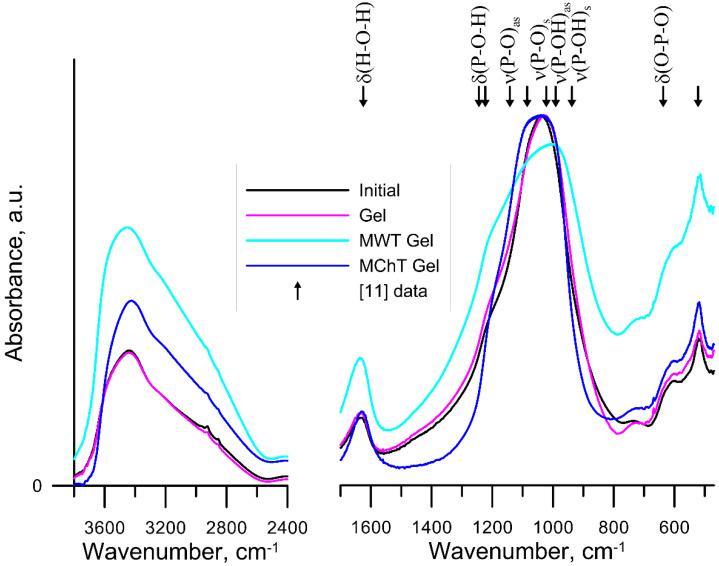
FTIR spectra for the initial sample (black) and that after HTT (violet), MWT (cyan), and MChT (blue) of the gel ZrP samples. [11] Xu, Y et al., 2020.

**Figure 3 materials-15-05050-f003:**
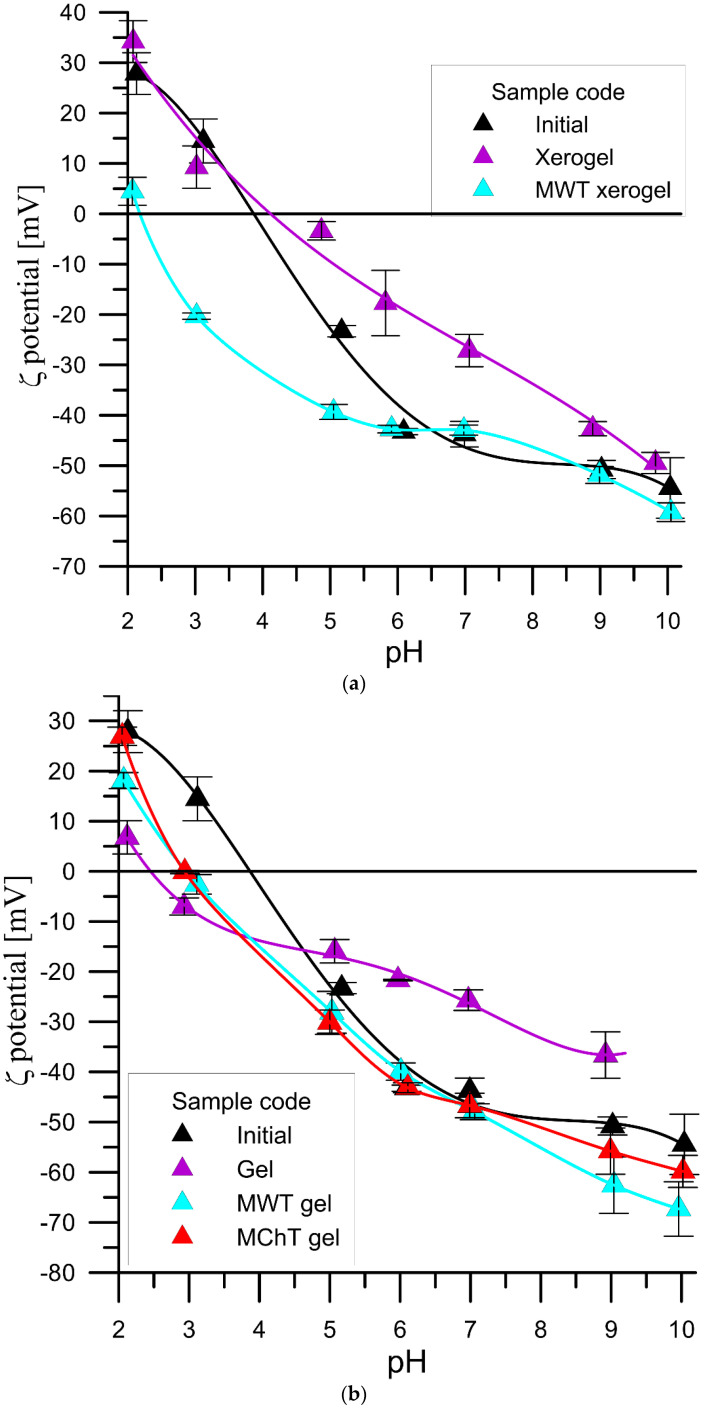
(**a**). Zeta potential of the zirconium phosphate xerogel samples MWT treated in the 0.001 mol/dm^3^ KNO_3_ solution as a function of pH. (**b**). Zeta potential of the zirconium phosphate gel samples MWT and MChT treated in the 0.001 mol/dm^3^ KNO_3_ solution as a function of pH.

**Figure 4 materials-15-05050-f004:**
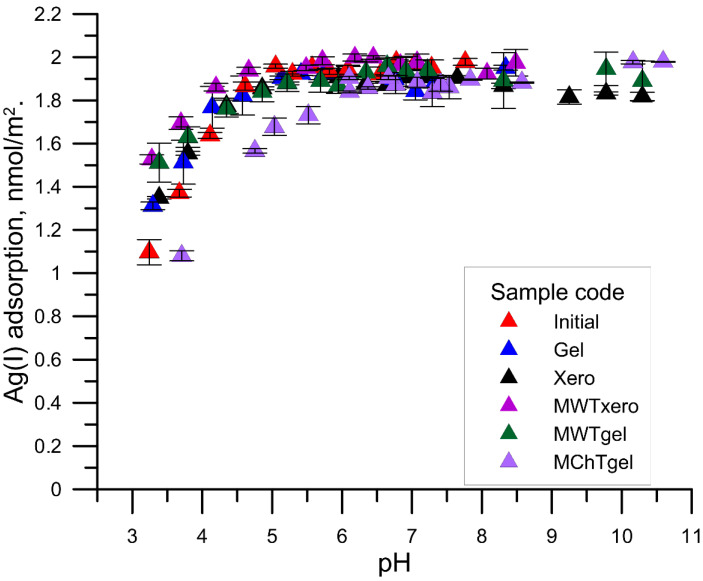
Comparison of Ag (I) ions adsorption in different zirconium phosphate samples from the solution of the initial concentration, 1 μmol/dm^3^ Ag (I) ions as a function of pH.

**Figure 5 materials-15-05050-f005:**
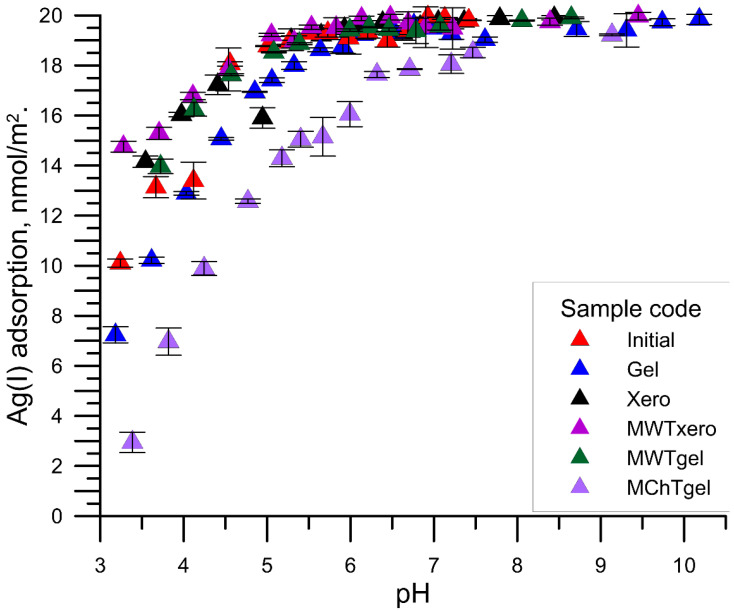
Comparison of Ag (I) ions adsorption in different zirconium phosphate samples from the solution of initial concentration, 10 μmol/dm^3^ Ag (I) ions as a function of pH.

**Figure 6 materials-15-05050-f006:**
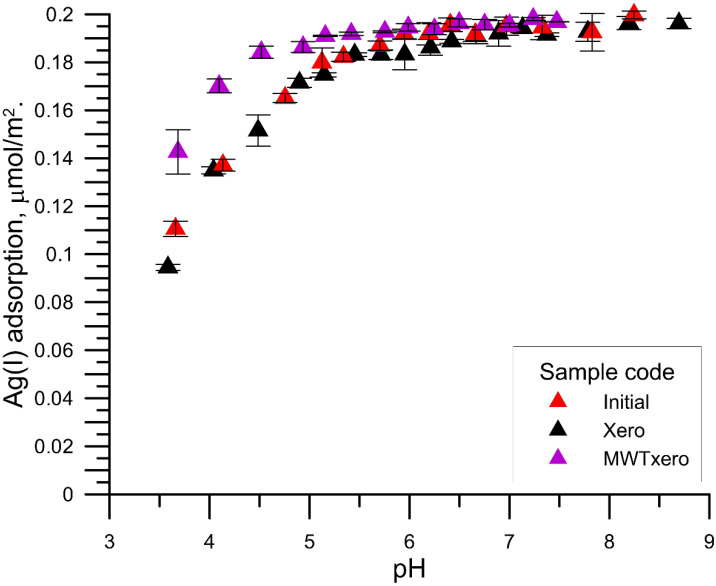
Comparison of Ag (I) ions adsorption in different zirconium phosphate samples from the solution of initial concentration, 100 μmol/dm^3^ Ag (I) ions as a function of pH.

**Table 1 materials-15-05050-t001:** The specific surface area of the zirconium phosphate samples, the weight of the sample used in the adsorption tests and the surface area of the sample.

Preparation Conditions	Sample Designation	Specific Surface Area (m^2^/g)	Adsorbent Mass (g)	Adsorbent Surface m^2^
Initial precipitated	Initial	195	0.1283	25.0
HTT 150 °C Xerogel	Xero	278	0.0899	25.0
HTT 150 °C Gel	Gel HTT	450	0.0556	25.0
MWT 230 °C xerogel	MWT Xero	234	0.1068	25.0
MWT 230 °C gel	MWT Gel	319	0.0784	25.0
MChT 500 rpm gel	MChT Gel	406	0.0616	25.0

**Table 2 materials-15-05050-t002:** pH_iep_ and p_Ka2_ of the surface groups of initial zirconium phosphate and after HTT, MWT, and MChT.

Label	pHiep	pKa1	pKa2
	0.001		
Xerogel			
Initial	3.9	0.87	6.28
Xerogel	4.1	0.52	7.24
MWT xerogel	2.2		4.70
Gel			
Initial	3.9	0.87	6.28
Gel	2.5		5.36
MWT gel	2.9		5.60
MChT gel	2.9		5.47

**Table 3 materials-15-05050-t003:** The equilibrium constants of Ag (I) adsorption and the amount of released H + ions based on the non-stoichiometric exchange model.

Sample No	Sample Code	Ag [I] Conc. (μmol/dm^3^)	p*β**_n_**^s^*	*n*	pH_50%_	∆pH_90–10%_
**1**	Initial	1	0.065	0.77	3.2	2.5
		10	0.033	0.66	3.3	2.9
		100	0.016	0.54	3.5	3.5
**2**	Xerogel	1	0.040	0.55	2.8	3.5
		10	0.044	0.56	2.9	3.4
		100	0.016	0.62	3.7	3.1
**3**	Gel	1	0.045	0.58	2.8	3.3
		10	0.016	0.57	3.6	3.3
**4**	MWT Xerogel	1	0.141	0.73	2.6	2.6
		10	0.034	0.49	2.7	3.9
		100	0.076	0.78	3.1	2.5
**5**	MWT gel	1	0.029	0.38	2.1	5.0
		10	0.029	0.55	3.1	3.5
**6**	MChT gel	1	0.011	0.36	3.1	5.3
		10	0.006	0.62	4.4	3.1

## Data Availability

Not applicable.

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
