# Peer review of "Adsorption of Ag (I) Ions at the Zirconium Phosphate/KNO3 Aqueous Solution Interface"

_materials, 2022, doi:10.3390/ma15145050_

Round 1

Reviewer 1 Report

The manuscript entitled: “Adsorption of Ag (I) ions at the zirconium phosphate/KNO3 2 aqueous solution interface” authored by Janusz et al, is an excellent study. The manuscript is rich, and it would be an added value to the scientific community of the Materials Journal.

I have a very minor comments/suggestions prior to acceptance for publication:  

1.       Abstract:

-          Major findings and numeric values shall be presented in the abstract.

2.       Introduction:

-          The novelty of this study is well-described. Please add few sentences to attract audience.

3.       Experimental procedure:

-          The authors referred to reference [17] for the characterization procedures. I would suggest adding a scheme showing the main tools used and the experimental conditions for the characterization.  

4.       Conclusions:  

-          How feasible is this zirconium phosphate adsorbent, especially when you have 3 different methods? How can you attract the industry to your adsorbent?

-          What is the future work in this domain?         

Author Response

Review Answer 1

  1. Abstract:

Major findings and numeric values shall be presented in the abstract.

The abstract was corrected as suggested.

  1. Introduction:

The novelty of this study is well-described. Please add few sentences to attract audience.

As can be seen from the presented results, the use of a short (only 0.5 hour) microwave treatment has a significant effect on the physicochemical characteristics and, ultimately, on the surface and adsorption properties of adsorbents. This is due to the combined action of hydrothermal conditions and a specific non-thermal effect. It should be noted that for such a common adsorbent as zirconium phosphate, this is a novelty.  Corresponding sentence was added to Conclusions.

  1. Experimental procedure:

The authors referred to reference [17] for the characterization procedures. I would suggest adding a scheme showing the main tools used and the experimental conditions for the characterization.

We have made additions to the text on p. 2

  1. Conclusions:  

How feasible is this zirconium phosphate adsorbent, especially when you have 3 different methods? How can you attract the industry to your adsorbent?

We believe that microwave method of modification is the most suitable. This method stands out from those studied, on the one hand, by its maximum effect and, on the other hand, by its shorter duration and lower energy

What is the future work in this domain?   

Testing of the antimicrobial activity of zirconium phosphate samples with silver deposited.

The English language has been checked.

Reviewer 2 Report

Comments

In this manuscript, different methods were applied for zirconium phosphate sample modification in order to improve their adsorption affinity for the Ag(I) ions. The Ag+ ion adsorption studies the MWT treatment of zirconium phosphate samples causing the greatest affinity of Ag+ ions. I don’t see any novelty in this work. Also, there is also lack of information which makes it difficult to completely understand developed materials. I do not recommend this manuscript. Following are some comments which can be addressed for improvement of this manuscript.

1-Is there any leaching from zirconium phosphate samples at lower pH? if yes,  

2-Novelty should be added for the present work.

3-What was the effect of the synthesis method on morphology surface area etc.?

Author Response

Review Answer 2

1-Is there any leaching from zirconium phosphate samples at lower pH? if yes,  

The advantage of zirconium phosphate is its chemical stability, it is insoluble in acid and alkaline solutionse.g. solutions (for example, Lee, J. Y., Vyas, C. K., Kim, B.-R., Kim, H. J., Hur, M. G., Yang, S. D., … Kim, S. W. (2016). Acid resistant zirconium phosphate for the long term application of 68 Ge/ 68 Ga generator system. Applied Radiation and Isotopes, 118, 343–349. doi:10.1016/j.apradiso.2016.09.0; W. Janusz, V. Sydorchuk, E. Skwarek, S. Khalameida, Effect of hydrothermal treatment of zirconium phosphate xerogel on its surface groups properties and affinity adsorption to Cd (II) ions from acidic solutions, Materials Research Bulletin 148 (2022) 111674. There is no information in the literature on the solubility product of zirconium phosphate.

2-Novelty should be added for the present work.

As can be seen from the presented results, the use of a short (only 0.5 hour) microwave treatment has a significant effect on the physicochemical characteristics and, ultimately, on the surface and adsorption properties of adsorbents. This is due to the combined action of hydrothermal conditions and a specific non-thermal effect. It should be noted that for such a common adsorbent as zirconium phosphate, this is a novelty.  Corresponding sentence was added to Conclusions

3-What was the effect of the synthesis method on morphology surface area etc.?

Work [17] is devoted to a detailed study of the influence of these modification methods on physico- chemical characteristics, including on morphology and specific surface of zirconium phosphate. In the presented work, the surface structure and adsorption were studied only for a part of the samples synthesized earlier.

The English language has been checked.

Reviewer 3 Report

The present work studies the adsorption of Ag(I) ions at the zirconium phosphate/KNO3 interface. For this purpose, the authors used the mechanical (MChT), microwave (MWT), and hydrothermal (HTT) methods of modification of adsorption affinity by Ag(I) ions. Overall, the work has great potential and is well-designed and well-written. However, below are some suggestions for improving the work:

1) line 26: "E. coli and S. aureus" - In its first appearance, write without abbreviations and always use the italic font (Escherichia coli and Staphylococcus aureus);

2) In the Introduction, a paragraph is needed emphasizing the importance and contribution of this work;

3) Figures 1-6 should be improved for publication;

4) The text needs polishing, as there are many grammatical and formatting errors;

5) The Conclusion should be improved as it is very concise.

Author Response

Review Answer 3

  • line 26: "E. coli and S. aureus" - In its first appearance, write without abbreviations and always use the italic font (Escherichia coli and Staphylococcus aureus);

In the new version of the manuscript, the names of the bacteria have been corrected.

  • In the Introduction, a paragraph is needed emphasizing the importance and contribution of this work;

The introduction is somewhat extended in order to show the importance of the work and its contribution to the study of the adsorption properties of phosphates.

3) Figures 1-6 should be improved for publication;

Fig 1-6 was prepared according to the instructions for the authors, i.e. with a resolution of a 600dpi, color RGB at 24-bit per channel and in TIFF format.

4) The text needs polishing, as there are many grammatical and formatting errors;

The errors in the text have been corrected.

5) The Conclusion should be improved as it is very concise.

- The Conclusions were supplemented with the main results and numeric values of important parameters...

The English language has been checked.

Round 2

Reviewer 2 Report

Comments

Now after revision, quality of this manuscript is improved. However, there are still need some minor corrections.

 1)    Check following text in the abstract and correct: Compared to the ZrP Initial sample, he observes the shift of the Ag (I) ion adsorption edge towards lower pH value.

 2)    Line no 53.  Correct “There-fore” as “Therefore”

 3) There are many typos which must be checked carefully. English language must be improved.

4)    Line no 237,  close the bracket for reference ”[27”

Author Response

Review Answer 2

  • Check following text in the abstract and correct: Compared to the ZrP Initial sample, he observes the shift of the Ag (I) ion adsorption edge towards lower pH value.

Has been corrected.

  • Line no 53.  Correct “There-fore” as “Therefore”

Has been corrected.

  • There are many typos which must be checked carefully. English language must be improved.

The English language has been checked by an English translator. Relevant corrections are marked in red in the text of the publication.

4)    Line no 237, close the bracket for reference ”[27”

Has been corrected.